# A short linear motif in scaffold Nup145C connects Y-complex with pre-assembled outer ring Nup82 complex

Roman Teimer[1], Jan Kosinski[2], Alexander von Appen[2], Martin Beck [2] & Ed Hurt[1]

Nucleocytoplasmic transport occurs through nuclear pore complexes (NPCs), which are formed from multiple copies of ~30 different nucleoporins (Nups) and inserted into the double nuclear membrane. Many of these Nups are organized into subcomplexes, of which the Y-shaped Nup84 complex is the major constituent of the nuclear and cytoplasmic rings. The Nup82–Nup159–Nsp1 complex is another module that, however, is only assembled into the cytoplasmic ring. By means of crosslinking mass spectrometry, biochemical reconstitution, and molecular modeling, we identified a short linear motif in the unstructured N-terminal region of *Chaetomium thermophilum* Nup145C, a subunit of the Y-complex, that is sufficient to recruit the Nup82 complex, but only in its assembled state. This finding points to a more general mechanism that short linear motifs in structural Nups can act as sensors to cooperatively connect pre-assembled NPC modules, thereby facilitating the formation and regulation of the higher-order NPC assembly.

[1] Biochemistry Center of Heidelberg University (BZH), Im Neuenheimer Feld 328, 69120 Heidelberg, Germany. [2] Structural and Computational Biology Unit, European Molecular Biology Laboratory (EMBL), Meyerhofstraße 1, 69117 Heidelberg, Germany. Correspondence and requests for materials should be addressed to M.B. (email: martin.beck@embl.de) or to E.H. (email: ed.hurt@bzh.uni-heidelberg.de)

Nuclear pore complexes (NPCs) are large multiprotein assemblies embedded into the nuclear envelope of eukaryotic cells, enabling and controlling the migration of large molecules between the nucleus and the cytoplasm. Despite their large size (~50–120 MDa, depending on the organism), NPCs are assembled from only ~30 different proteins, termed nucleoporins (Nups), which are mostly conserved and present in multiple copies (usually 8–64) per NPC[1–3]. Specific sets of Nups interact with each other to form distinct, biochemically stable subcomplexes, which in vivo are embedded in a higher-order NPC network of octagonal symmetry[4]. The three major sub-structures of the NPC are the cytoplasmic, inner, and nuclear rings (CR, IR, and NR, respectively), which are stacked co-axially and build up the conserved core scaffold of the NPC, which is symmetric across the nuclear envelope and contains a central transport channel[5–7]. A subset of Nups is tethered to the IR and establishes the permeability barrier of the NPC by projecting phenylalanine–glycine (FG)-rich, intrinsically disordered regions into the central transport channel[8–12]. The nuclear basket and the cytoplasmic filaments are less conserved, peripheral substructures that protrude from the NR and the CR toward the nucleoplasm and the cytoplasm, respectively.

The best-characterized NPC subcomplex is the Y-shaped complex, so called because of its peculiar outline that is con-served across various species[6, 13–15]. In Saccharomyces cerevisiae (Sc) the Y-shaped complex (Nup84 complex) consists of seven protein members, namely Nup145C, Nup120, Nup85, Nup84, Nup133, Sec13, and Seh1[13, 16, 17]. The orthologous counterpart in vertebrates is the Nup107–Nup160 complex and contains three additional constituents—Nup37, Nup43, and ELYS[18, 19]. The superior biophysical properties of proteins from the thermophilic ascomycete Chaetomium thermophilum (Ct) have facilitated many structural studies of various macromolecular assemblies[20], including Nups[8–10, 15, 21–23]. The Nup84 complexes of C. ther-mophilum and other fungi also contain Nup37 and ELYS, but no Nup43 homolog[15, 24, 25]. Nup145C and Nup85 interact directly with Nup120, thus forming the central vertex of the characteristic Y structure (see also Figs. 1a and 2a). Nup145C recruits Sec13 and the elongated Nup84–Nup133 dimer, while Nup120 recruits the Nup37–ELYS module in the case of C. thermophilum. In yeast and other organisms Seh1 is attached to the complex via Nup85, whereas this interaction is absent in thermophilic ascomycetes such as C. thermophilum[15, 26]. Furthermore, the Y-shaped com-plex can dimerize in vitro and further oligomerize in a head-to-tail manner to form the NPC's outer rings, each containing 16 copies of the complex[6, 15].

Various data suggest that the Y-shaped complex serves as an attachment site for the asymmetrical features of the NPC. In the yeast S. cerevisiae, for instance, Nup145C is required for the docking of Mlp1 and Mlp2, the main constituents of the nuclear basket, whereas Nup60, another basket protein, is attached via Nup84[27, 28]. Consistently, a proximity-dependent labeling study implicated that the Nup107–Nup160 complex anchors TPR, the vertebrate homolog of Mlp1/Mlp2, at the NR[29]. In contrast, the vertebrate Nup358–RanGAP1*SUMO1–Ubc9 complex is exclu-sively located on the cytoplasmic face of the NPC and knockdown of Nup358 results in the loss of 50% of the Y complexes in the CR[30, 31].

The conserved Nup82 complex (Nup82–Nup159–Nsp1 in S. cerevisiae and C. thermophilum, Nup88–Nup214–Nup62 in ver-tebrates) is another NPC module that localizes exclusively to the cytoplasmic side of the NPC as part of the cytoplasmic filaments[32, 33]. The ScNup82 complex contains an additional subunit, Dyn2, that promotes the dimerization of the complex both in vitro and in vivo[34]. In yeast, the Nup82 complex is functionally connected to mRNA export, as it tethers the RNA helicase Dbp5 to the NPC, which contributes to the dismantling of messenger ribonucleoproteins after nuclear export[35, 36]. Several studies indicated that the ScNup82 complex binds close to the ScNup85–Seh1 arm of the Y, but the details of this interaction have not been revealed[6, 34, 37]. Biochemical studies suggest that the Nup82 complex is anchored to the NPC by interacting with Nup145N (and/or its additional yeast homologs Nup100 and Nup116)[8, 38–40]. Also, Nup145C might contribute to anchoring the Nup82 complex[41]. Nup145C, together with Nup145N, emerges from the common Nup145 precursor protein by co-translational autoproteolytic cleavage[42–46]. Whereas Nup145N is associated with the IR, Nup145C, as part of the Y-shaped com-plex, is located at the outer (both nuclear and cytoplasmic) rings, raising the question of whether the two Nup82 complex tethering mechanisms are of equal functional relevance in the assembled NPC. A detailed analysis of the exact interaction sites of the respective Nups would therefore be invaluable to address this question.

To characterize the interaction between the CtNup82 complex and the CtY-shaped complex, we performed crosslinking mass spectrometry (XL-MS) of the in vitro reconstituted CtNup82–Y supercomplex. Guided by the XL-MS analysis and multiple sequence alignments, we identified a short linear motif (SLiM) within the N-terminal domain (NTD) of CtNup145C that is sufficient to recruit the assembled CtNup82 complex, but not its individual subunits. We found this mechanism of cooperative binding to be similar to the interaction between the interaction motif 1 (IM-1) of IR nucleoporin Nic96 and the assembled Nsp1 channel heterotrimer[8], which was recently crystallized[9]. Based on homology to this complex, the distance restraints obtained by XL-MS, and the biochemical data, we propose how the SLiM of CtNup145C might interact with the CtNup82 complex.

## Results

**Y- and CtNup82 complex interact via CtNup145C-NTD.** The mechanisms of how the nucleoporin modules in the CR of the NPC are physically connected to each other have remained lar-gely unknown. We have recently observed a robust biochemical interaction between the C. thermophilum Nup82 complex and the Y-shaped Nup84 complex, in which the Y-subunit Nup145C was found to be involved in generating a contact between the two modules[41]. To gain insight into the molecular mechanism of this interaction, we reconstituted a supercomplex between the Y- and Nup82 complexes, and performed XL-MS. For this purpose, CtNup82–Nup159C–Nsp1C and a minimal Y-complex, the het-erotrimeric CtNup145C–Nup85–Nup120 (called the Y-vertex, Fig. 1a), were co-assembled and the derived CtNup82–Y supercomplex (CtNup82–Nup159C–Nsp1C–Nup145C–Nup85–Nup120) was analyzed by XL-MS using the bivalent crosslinking reagent disuccinimidyl suberate[6, 15]. This analysis revealed a large number of crosslinks between the subunits of the CtNup82 complex, similar to what has been found for the yeast Nup82 complex[34, 37]. Moreover, crosslinks within the Y-complex were observed, in particular between the C-terminal domain of CtNup120 and two regions in CtNup145C, one around residue K266 and the other in the C-terminal end. This is consistent with X-ray data of the yeast Y-shaped complex, in which the C-terminal domains of Nup120 and Nup145C are in direct con-tact[14] (Fig. 2a). The crosslink pattern of the Y complex, either alone[15] or in association with the Nup82 complex, was highly similar, suggesting no major conformational rearrangement occurs upon supercomplex formation.

In addition to the internal crosslinks within the two modules, we observed crosslinks between all subunits of the CtNup82 complex to members of the Y-complex, suggesting that both

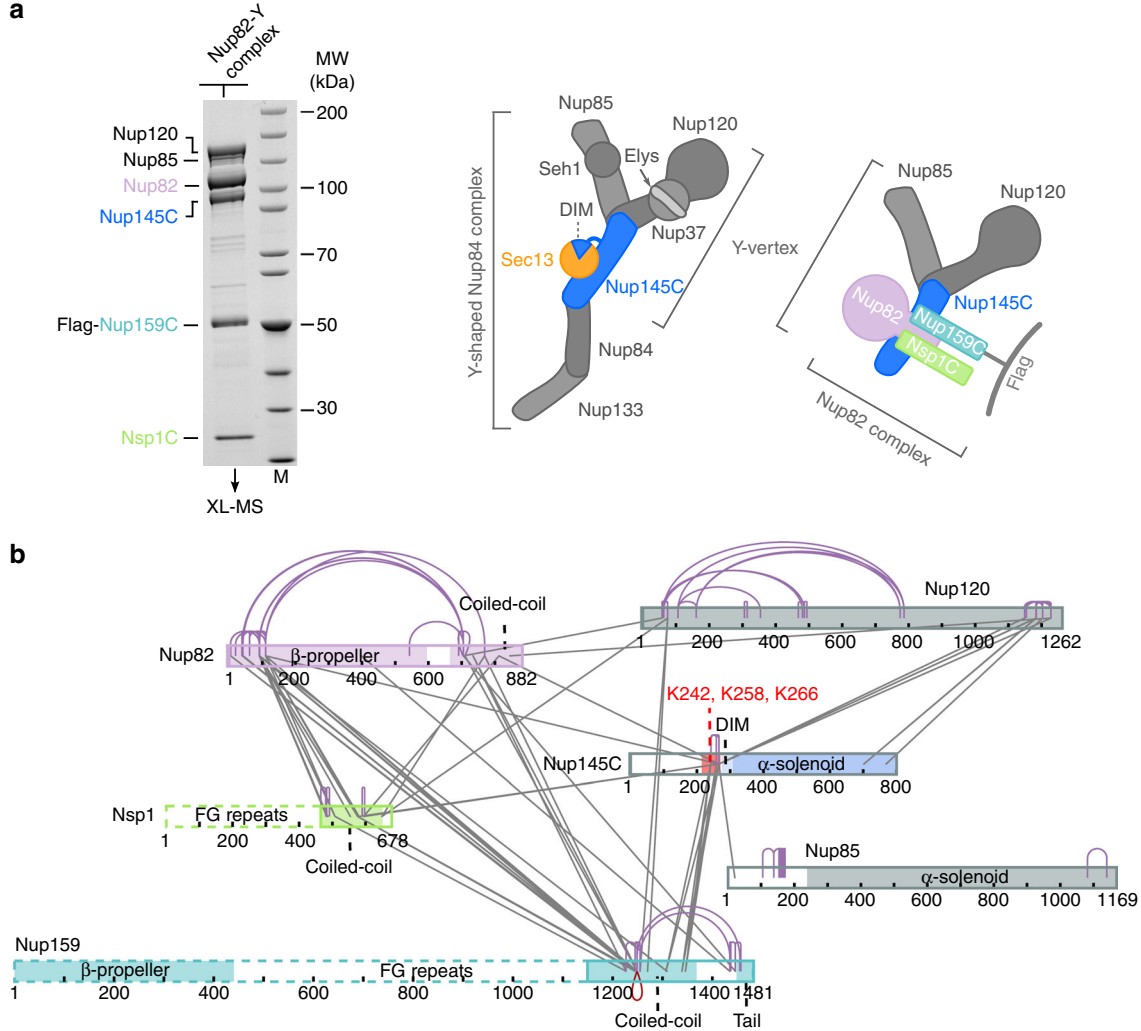

**Fig. 1** XL-MS analysis of the *Ct*Nup82 complex with the central vertex of the Y-shaped *Ct*Nup84 complex. **a** In vitro reconstitution of the *Ct*Nup82–Y complex (*Ct*Nup82–Nup159C–Nsp1C–Nup145C–Nup85–Nup120). An immobilized Flag-*Ct*Nup159C–Nup82–Nsp1C complex was incubated with the soluble *Ct*Nup145C–Nup85–Nup120 complex (Y-vertex). Shown is the analysis of the Flag eluate by SDS-PAGE and coomassie staining. *Ct*Nup145C is highlighted in blue as it interacts directly with the *Ct*Nup82 complex[41]. M marker, MW molecular weight. An uncropped image of the gel is shown in Supplementary Fig. 4a. **b** Schematic representation of the *Ct*Nup82–Y complex showing the crosslinks determined by mass spectrometry. Crosslinks are depicted as straight gray lines, intramolecular self-links as curved purple lines, and intermolecular self-links as dark red loops. DIM domain invasion motif. Additional data are shown in Supplementary Table 2 and Supplementary Data 1

modules come into close proximity to interact. Strikingly, most of these intermodule crosslinks are clustered in a 'hotspot' region of *Ct*Nup145C (residues 215–270), consistent with the biochemical data that identifies *Ct*Nup145C as the subunit that directly binds to the *Ct*Nup82 complex[41]. Within this *Ct*Nup145C hotspot located between the N terminus and the highly structured C-terminal α-solenoid domain, three neighboring lysine residues, K242, K258, and K266, were crosslinked to all subunits of the *Ct*Nup82 complex, and in particular to the coiled-coil domains of *Ct*Nup82, *Ct*Nsp1, and *Ct*Nup159 (Fig. 1b). These data suggest that a distinct motif located between the N and C-terminal domains of Nup145C interacts with the heterotrimeric coiled-coil Nup82 complex.

Because no crystal structure of the *C. thermophilum* Y-complex is available, we looked for the equivalent hotspot residues in yeast Nup145C, based on the crystal structure of the yeast Nup84 complex[14] (Fig. 2a). Accordingly, *Sc*Nup145C consists of a largely unstructured NTD (residues 1–148), a short domain invasion motif (DIM, 149–183), followed by a structured α-helical

C-terminal domain (CTD, 184–711; Fig. 2b). Whereas the α-helical solenoid interacts with Nup120 and Nup84, the DIM recruits Sec13 by providing a missing β-sheet in trans to complete the seven-bladed β-propeller of Sec13 (Fig. 2a). Except for a short α-helix (approximate residues 92–99), which forms a contact with Nup85 (Fig. 2a), the rest of the N-terminal extension of Nup145C upstream of the DIM is unresolved in the crystal structure, suggesting that it is unstructured and/or flexible[14].

To test whether the NTD of *Ct*Nup145C is responsible for recruiting the *Ct*Nup82 complex, in vitro binding assays were carried out with bead-immobilized full-length *Ct*Nup145C and two truncated constructs, the N-terminal *Ct*Nup145C-NTD and the C-terminal *Ct*Nup145C-CTD (Fig. 2b, c). This analysis revealed that full-length *Ct*Nup145C was able to bind *Ct*Sec13 (via the DIM motif) as well as the isolated recombinant *Ct*Nup82 complex, either separately or both simultaneously (Fig. 2c, lanes 3–6). In contrast, *Ct*Nup145C-CTD no longer recruited the *Ct*Nup82 complex, but was still able to bind *Ct*Sec13 (Fig. 2c, lanes 7–10). *Ct*Nup145C-NTD was sufficient to efficiently bind

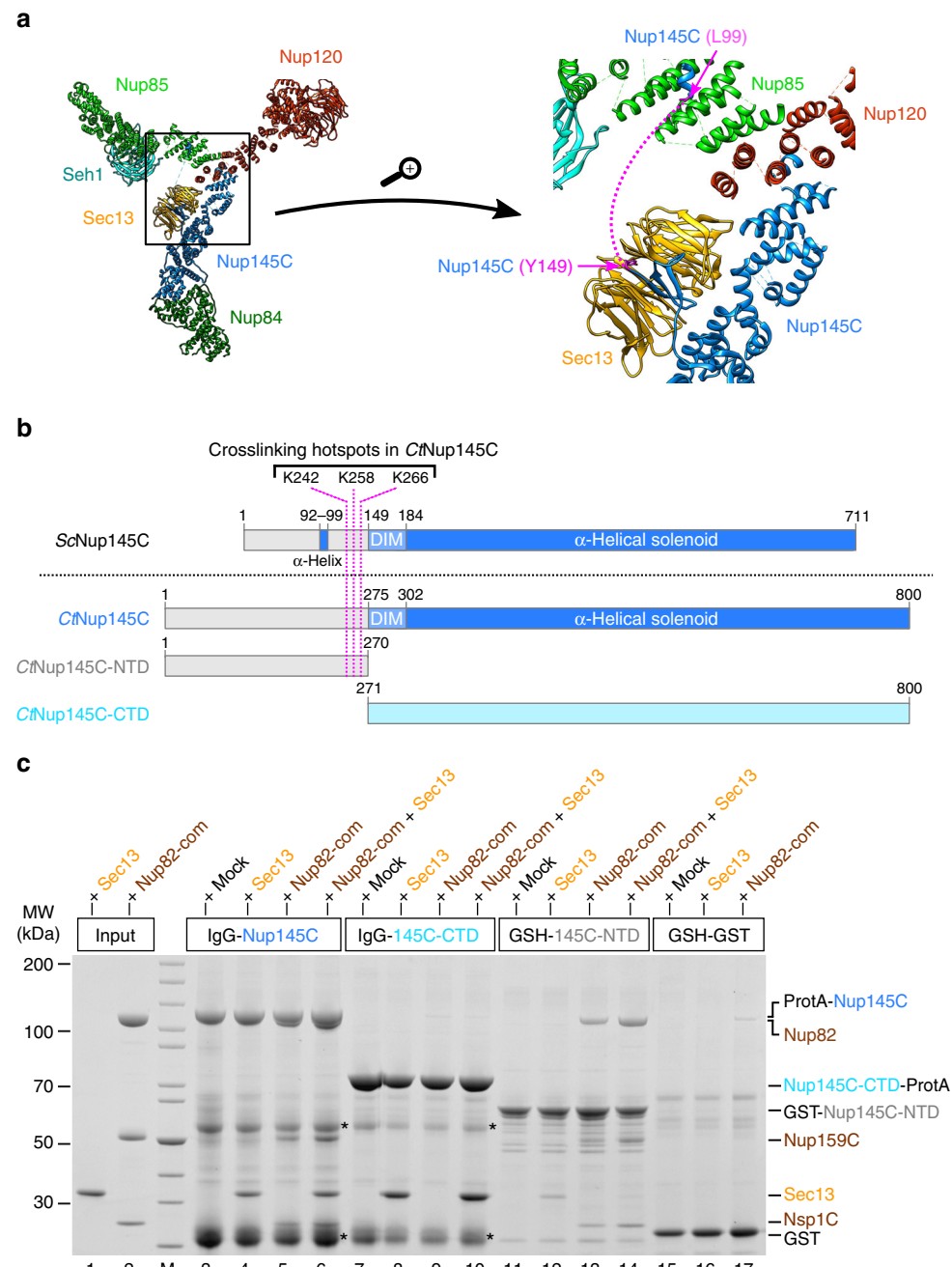

**Fig. 2** The N-terminal domain (NTD) of *Ct*Nup145C is both necessary and sufficient for the recruitment of the *Ct*Nup82 complex. **a** Crystal structure of the majority of the Y-shaped *Sc*Nup84 complex (PDB ID: 4xmm). The region upstream of the *Sc*Nup145C-DIM domain is indicated by a dashed pink line (right panel) as it is missing from the crystal structure. **b** Schematic representation of *Sc*Nup145C and *Ct*Nup145C and the design of truncated *Ct*Nup145C-NTD and *Ct*Nup145C-CTD; see also Supplementary Fig. 2. The crosslinking hotspots found in *Ct*Nup145C-NTD (Fig. 1b) are indicated by dashed pink lines. DIM domain invasion motif. **c** In vitro binding assay with immobilized *Ct*Nup145C, *Ct*Nup145C-NTD, or *Ct*Nup145C-CTD and soluble *Ct*Sec13 and/or the *Ct*Nup82–Nup159C–Nsp1C complex (Nup82-com). Shown is the analysis of SDS eluates by SDS-PAGE and coomassie staining. The experiment was performed at least twice with consistent results. Asterisks indicate bands corresponding to IgG heavy and light chains; M marker, Mock purification buffer with *E. coli* whole cell lysate, MW molecular weight. An uncropped image of the gel is shown in Supplementary Fig. 4b

the *Ct*Nup82 complex, but only a small amount of *Ct*Sec13 was co-enriched (Fig. 2c, lanes 11–14). We attribute this latter binding to be unspecific as Sec13 probably adopts an incomplete β-propeller fold in the absence of its binding partner Nup145C-DIM. Together, these data indicate that the NTD of *Ct*Nup145C is both necessary and sufficient to recruit the *Ct*Nup82 complex.

To further extend these studies, the reconstituted *Ct*Nup82 complex was immobilized on beads and tested for binding to

*Ct*Sec13, *Ct*Nup145C, or the minimized Y-vertex (*Ct*Nup145C–Nup85–Nup120) as described above (Fig. 3). As anticipated, the *Ct*Nup82 complex recruited full-length *Ct*Nup145C but only trace amounts of the construct lacking the N-terminal extension (i.e., *Ct*Nup145C-CTD, Fig. 3, lanes 8 and 9). The fact that *Ct*Nup145C-CTD did not bind to the negative control (Fig. 3, lane 19) points towards a low-affinity interaction between *Ct*Nup145C-CTD and the *Ct*Nup82 complex.

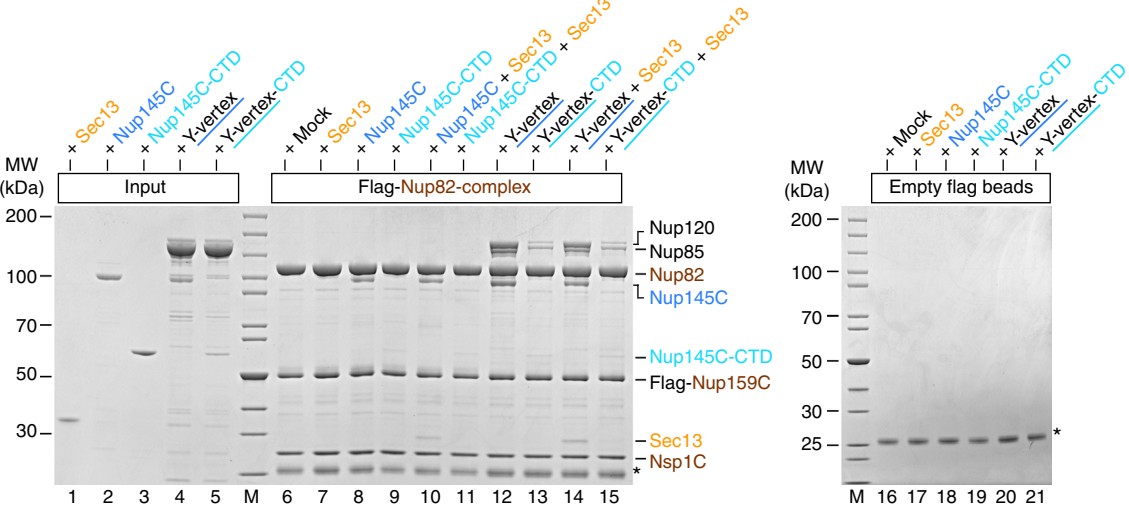

**Fig. 3** Interaction between the *Ct*Nup82 complex and the *Ct*Y-vertex. In vitro binding assay with the immobilized *Ct*Nup82 complex (Flag-*Ct*Nup159C–Nup82–Nsp1C) and different soluble prey proteins of the *Ct*Y-shaped complex. Shown is the analysis of SDS eluates by SDS-PAGE and coomassie staining. The experiment was performed at least twice with consistent results. The asterisk indicates bands corresponding to the anti-Flag light chain; *M* marker, Mock purification buffer with *E. coli* whole cell lysate, MW molecular weight, Y-vertex co-purified *Ct*Nup145C–Nup85–Nup120 complex, Y-vertex-CTD co-purified *Ct*Nup145C-CTD–Nup85–Nup120 complex. Uncropped images of the gels are shown in Supplementary Fig. 4c, d

Consistently, the *Ct*Nup82 complex efficiently recruited the Y-vertex, but only if it contained the full-length *Ct*Nup145C (Fig. 3, lane 12). In contrast, the Y-vertex carrying the truncated *Ct*Nup145C-CTD was only inefficiently recruited to the bead-immobilized *Ct*Nup82 complex (Fig. 3, lanes 13 and 15). The observed residual interaction is consistent with the aforementioned low affinity of *Ct*Nup145C-CTD towards the *Ct*Nup82 complex and with previous findings[41], which might imply that other members of the Y-complex (e.g., Nup85) also contribute to the overall binding of the Nup82 complex, as previously reported in the yeast system[37]. Finally, when the *Ct*Nup82 complex was incubated with the Y-vertex in the presence of *Ct*Sec13, an assembly of seven subunits, *Ct*Nup82–Nup159C–Nsp1C–Nup145C–Sec13–Nup85–Nup120, was reconstituted (Fig. 3, lane 14).

Based on the data from *S. cerevisiae*, we performed reconstitution studies using recombinant yeast Y-complex and purified yeast Nup82 complex (i.e., immobilized *Sc*Y-complex: GST-*Sc*Nup145C–Sec13–Nup120–Nup85–Seh1; added soluble *Sc*Nup82-complex: *Sc*Nup159-ΔFG–Nsp1C–Nup82–Dyn2). However, we did not find a significant interaction between Y-complex and Nup82 complex from yeast under our standard in vitro binding conditions established for the *C. thermophilum* Nups (Supplementary Fig. 1). It is possible that the interaction between *C. thermophilum* Y-complex and Nup82 complex is more robust in vitro than that between the yeast Y- and Nup82 complexes, but it is also conceivable that there are differences between organisms (see Discussion). Thus, it is currently not possible to make a direct comparison between our study and that of Fernandez-Martinez and colleagues[37] regarding the mechanisms of how Y-complex and Nup82 complex interact on a molecular basis.

**A SLiM in *Ct*Nup145C-NTD recruits the *Ct*Nup82 complex**. To reveal the mechanism by which Nup145C bridges the Y- and Nup82 complexes, we searched for motifs in the N-terminal extension of Nup145C that might possibly mediate such an interaction. Multiple sequence alignment of Nup145C orthologs from distantly related organisms such as *C. thermophilum*, *S. cerevisiae*, *Xenopus laevis*, and *Homo sapiens* (Nup96 in humans)

showed that the N-terminal extension is not strongly conserved between these species. However, restricted alignment of the orthologs from different clades such as Pezizomycotina (of which *C. thermophilum* is member), Saccharomycotina (of which *S. cerevisiae* is member), and Metazoa (of which *Homo sapiens* is member), revealed distinct conserved blocks, which might represent SLiMs with the potential to bind to structured Nups as previously observed[8, 21]. Interestingly, one highly conserved motif in *Ct*Nup145C orthologs of the Pezizomycotina clade (called *Ct*Nup145C-B) directly preceding the Sec13-recruiting DIM contains the residues K242, K258, and K266 that were specifically crosslinked to the *Ct*Nup82 complex (see Fig. 1 and Supplementary Fig. 2a). Although the conserved motifs from the Saccharomycotina or metazoan clades are not sufficiently similar to confidently align them to the *Ct*Nup145C-B motif, a functionally related motif, with divergent sequence, might be also present in other Nup145C orthologs (Supplementary Fig. 2).

Based on this sequence alignment and insight from the X-ray structure, we performed in vitro binding assays using Flag-tagged *Ct*Nup145C-B as a bait. Strikingly, *Ct*Nup145C-B efficiently recruited the *Ct*Nup82 complex, whereas another sequence upstream of *Ct*Nup145C-B (*Ct*Nup145C-A, residues 163–195; see Supplementary Fig. 2a) was inert for such binding (Fig. 4b, compare lanes 10 and 6, respectively). Vice versa, the bead-immobilized *Ct*Nup82 complex effectively bound to purified, GST-labeled *Ct*Nup145C-B, but not to GST-*Ct*Nup145C-A (Fig. 4b, lanes 14–16). Thus, motif "B" in Nup145C is a bona fide "Nup82-complex interaction motif" (termed 82CIM). Apparently, the short 82CIM motif binds very stably to the *Ct*Nup82 complex, indicated by the fact that in vitro reconstituted *Ct*Nup82–Nup159C–Nsp1C–Nup145C-B complex did not dissociate during gel filtration chromatography (Supplementary Fig. 3). Accordingly, the *Ct*Nup145C-B elution peaked several fractions earlier when in complex with the *Ct*Nup82 complex as compared to purified *Ct*Nup145C-B alone, indicating a stable integration of *Ct*Nup145C-B into the *Ct*Nup82 complex. These data show that the critical motif to recruit the Nup82 complex to the Y-complex resides in a relatively short and conserved sequence, embedded in the flexible N-terminal extension of *Ct*Nup145C.

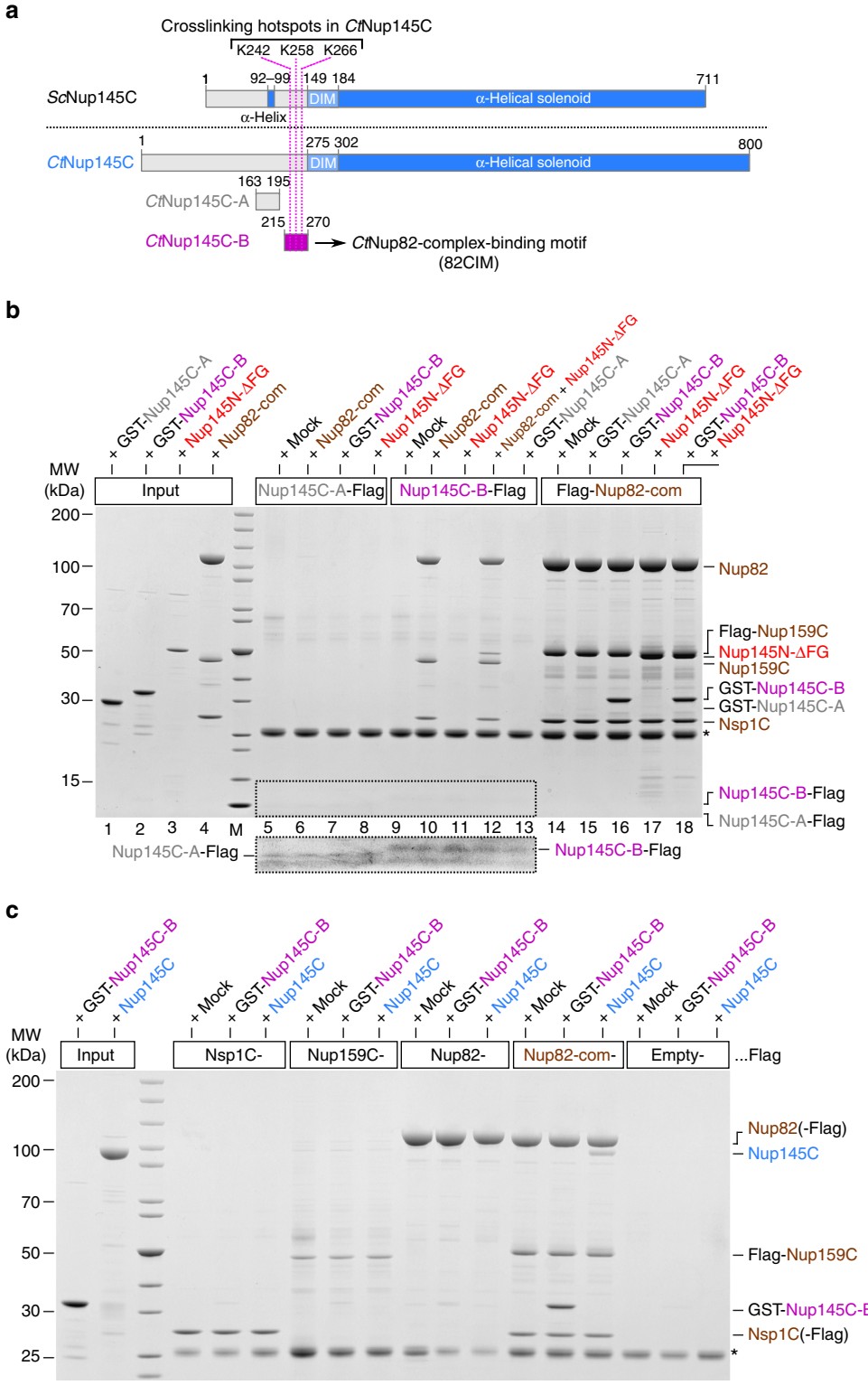

**Fig. 4** Identification of a short linear motif (SLiM) in *Ct*Nup145C-NTD that recruits the *Ct*Nup82 complex. **a** Schematic representation of *Sc*Nup145C and *Ct*Nup145C and the design of truncated *Ct*Nup145C-A and *Ct*Nup145C-B motifs; see also Supplementary Fig. 2. The crosslinking hotspots found in *Ct*Nup145C-NTD (Fig. 1b) are indicated by dashed pink lines. DIM domain invasion motif. **b** In vitro binding assay with Flag-immobilized *Ct*Nup145C-A, *Ct*Nup145C-B, and *Ct*Nup82 complex (Flag-*Ct*Nup159C–Nup82–Nsp1C) and soluble GST-*Ct*Nup145C-A, GST-*Ct*Nup145C-B, *Ct*Nup82 complex, and/or *Ct*Nup145N-ΔFG. Shown is the analysis of SDS eluates by SDS-PAGE and coomassie staining. The image section marked by the dashed box was subjected to image processing to increase the visibility of the small bait peptides *Ct*Nup145C-A-Flag and *Ct*Nup145C-B-Flag. **c** In vitro binding assay with Flag-immobilized *Ct*Nsp1C, *Ct*Nup159C, *Ct*Nup82, and *Ct*Nup82 complex (Flag-*Ct*Nup159C–Nup82–Nsp1C) and soluble GST-*Ct*Nup145C-B or *Ct*Nup145C. The experiments were performed at least twice with consistent results. The asterisk indicates bands corresponding to the anti-Flag light chain; M marker, Mock purification buffer with *E. coli* whole cell lysate, MW molecular weight. Uncropped gel images are shown in Supplementary Fig. 4e, f

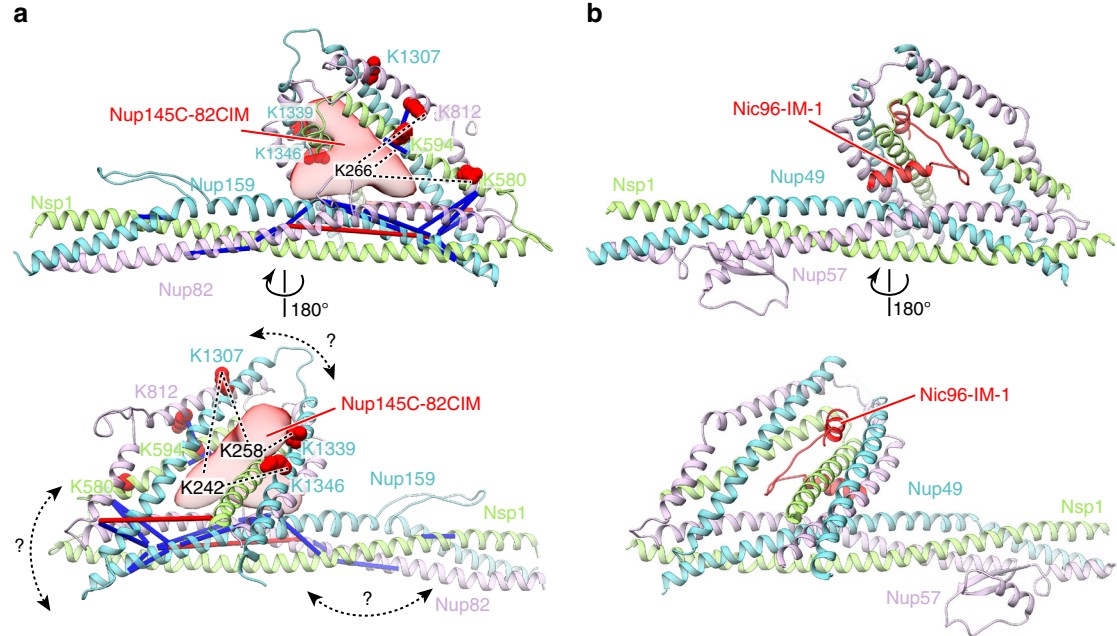

**Fig. 5** XL-MS suggests that *Ct*Nup145C binding to the *Ct*Nup82 complex resembles *Ct*Nic96 binding to the *Ct*Nsp1 complex. The model of the *Ct*Nup82 complex **a** was built based on the *Ct*Nsp1 complex (**b**, PDB ID: 5cws). The *Ct*Nup145C-82CIM motif (residues 215–270), which could not be reliably modeled at the atomic level, is shown as a low-resolution density. Note that although sequence similarity suggests that the *Ct*Nup82 complex has an architecture similar to the *Ct*Nsp1 complex; the exact orientation of the coiled-coil domains might be different (curved double arrows). Residues crosslinked to *Ct*Nup145C-82CIM are shown as red spheres and the crosslinked residues of *Ct*Nup145C are marked as K242, K258, and K266. Crosslinks between *Ct*Nup82, *Ct*Nsp1, and *Ct*Nup159 are indicated as sticks, crosslinks to *Ct*Nup145C as dashed lines. Crosslinks that satisfy the distance threshold of 30 Å are colored blue, and red otherwise. The two violated (red) crosslinks (*Ct*Nup159 Lys1249-*Ct*Nup82 Lys710 and *Ct*Nsp1 Lys649-*Ct*Nup82 Lys769) have relatively low ld-scores of 25.3 and 28.46, respectively. They thus might be false positives, in line with the calibrated FDR of 5% for the entire XL-MS data set. Alternatively, flexibility or alternative arrangements of coiled-coil domains might account for the distance violations

***Ct*Nup145N and -C bind simultaneously to the *Ct*Nup82 complex**. Based on our previous findings that the *Ct*Nup82 complex can also bind to *Ct*Nup145N, a linker Nup that, via its autoproteolytic domain, interacts with the β-propeller of Nup82[8], we sought to test whether further recruitment of Nup145N to the Nup82–Y supercomplex is possible. For this purpose, we tested the binding of a *Ct*Nup145N that is devoid of FG repeats (*Ct*Nup145N-ΔFG, residues 606–993) to immobilized *Ct*Nup145C-B in the absence and presence of the *Ct*Nup82 complex. Evidently, *Ct*Nup145N-ΔFG was only bound to immobilized *Ct*Nup145C-B when the *Ct*Nup82 complex was co-added, suggesting that the latter can simultaneously bind *Ct*Nup145N and *Ct*Nup145C (Fig. 4b, lanes 11 and 12). Thus, *Ct*Nup145N and *Ct*Nup145C are able to interact at the same time with the *Ct*Nup82 complex, suggesting that different regions of the Nup82 complex participate in these contacts.

**SLiM *Ct*Nup145C-B binds only pre-assembled *Ct*Nup82 complex**. To find out which subunit(s) of the Nup82 complex assist in binding the short linear *Ct*Nup145C-B motif, we tested the individual members of the *Ct*Nup82 complex, *Ct*Nsp1C, *Ct*Nup159C, and *Ct*Nup82, as baits in an in vitro binding assay (Fig. 4c). Strikingly, only the pre-assembled *Ct*Nup82 complex, but none of the immobilized individual subunits, could bind full-length *Ct*Nup145C or its derived minimal *Ct*Nup145C-B construct (Fig. 4c). Thus, it appears that upon assembly of the *Ct*Nup82 complex, a binding site is created that is crucial for interaction with the SLiM *Ct*Nup145C-B. Such cooperative binding is reminiscent of the SLiM IM-1 in Nic96, which only binds to the assembled heterotrimeric Nsp1 channel complex (*Ct*Nsp1–Nup49–Nup57), but not its single subunits[8].

Notably, it has been recently reported that the Nsp1–Nup49–Nup57 complex could be structurally related to the Nup82–Nup159–Nsp1 complex, as both complexes adopt a triple coiled-coil architecture[37]. Homology modeling of the *Ct*Nup82 complex based on the published crystal structure of the *Ct*Nsp1 complex in complex with the IM-1 of *Ct*Nic96[9] revealed that *Ct*Nup145C-B can bind similarly to the IM-1 (Fig. 5). Indeed, residues K242, K258, and K266 of the Nup145C-B motif crosslink exclusively to a region equivalent to the IM-1-binding site. The exact tertiary structure of the Nup82 and Nsp1 complexes may exhibit some differences since we could not find sequence similarity between *Ct*Nup145C-B and *Ct*Nic96-IM-1. Moreover, the coiled-coil domains likely adopt different arrangements[37]. Nevertheless, the cooperative binding mechanism and the XL-MS data strongly suggest that the triple coiled-coil domains of the *Ct*Nup82 complex form a composite binding site for the *Ct*Nup145C-B motif.

## Discussion

Although the overall structure and the architectural details of the NPC symmetric core and its connections is beginning to be understood, little is known about how peripheral and asymmetrically located modules are integrated into the NPC scaffold. To gain insight into these mechanisms, we focused on a physical interaction between the Y- and Nup82 complexes, as a paradigm of how modules of the outer CR are connected to each other. Based on *C. thermophilum* Nups, we reconstituted a supercomplex between the Y- and Nup82 modules and performed XL-MS to gain insight into the interaction interfaces between these subcomplexes. We found many crosslinks within the *Ct*Nup82 complex, indicating the predicted tightly packed coiled-coil

interaction between its subunits, and in good agreement with recent XL-MS studies in yeast and the detailed structural prediction of the *Sc*Nup82 complex obtained by integrative modeling[34, 37]. Moreover, the crosslinks within the Y-vertex of the *Ct*Nup82–Y supercomplex were not different to those obtained for the Y-vertex alone[15], suggesting that the central triskelion structure of the Y-complex does not undergo major structural rearrangements upon interaction with the Nup82 complex.

Most of the crosslinks between the *Ct*Nup82 complex and the *Ct*Y-vertex originated from the unstructured NTD of *Ct*Nup145C. By performing in vitro binding studies we could show that this region, in particular SLiM *Ct*Nup145-B, is sufficient to interact with the *Ct*Nup82 complex. This finding further supports an emerging concept that short unstructured domains found in linker Nups provide docking sites for other nucleoporins or subcomplexes, linking them together[8, 21]. Based on multiple sequence alignment, the *Ct*Nup145-B motif is moderately conserved between distant organisms on the basis of the primary structure (Supplementary Fig. 2b, c). Although both yeast and animals harbor conserved motifs in the N-termini of *Sc*Nup145C and Nup96 respectively, their similarity to *Ct*Nup145-B is not significant. It is therefore not clear whether a related interaction between the Y- and Nup82 complexes occurs in yeast or humans. It cannot be excluded that additional contacts between Nup85 and Nup82 such as proposed for yeast[37] may play a role in these organisms. Nonetheless, the yeast Nup82 complex was shown to become tethered to the NPC via Nup145N, as well as via its homologs Nup116 and Nup100[8, 38–40].

The interaction between Y- and Nup82 complex in *C. thermophilum* is consistent with the recent, aforementioned study done in yeast implying that another member of the Y-complex, Nup85, significantly contributes to the overall binding of the Nup82 complex[37]. This notion was based on that Nup85 and Nup145C crosslink to members of the Nup82 complex and that truncations of the Nup85/Seh1 arm affect the incorporation of Nup82 into the NPC. However, no in vitro reconstitution was performed or stoichiometric binding observed. Since our in vitro reconstitution studies with yeast Y- and Nup82 complexes were negative, we cannot further address the question of how yeast Y-Nups directly contribute to binding to the yeast Nup82 complex, as suggested by Fernandez-Martinez and coworkers[37]. Notably, the SLiM as found in *Ct*Nup145C is either absent or has a divergent sequence in yeast Nup145C (Supplementary Fig. 2b) suggesting that the *Sc*Nup145C could contribute less to this interaction between the yeast Y- and Nup82 complexes. Consistent with this data, deletion of the entire NTD from yeast Nup145C, which includes hypothetical SLiMs, did not cause a growth defect in a yeast *nup145* null mutant in the presence of plasmid-borne Nup145N, making it unlikely that the targeting of the yeast Nup82 complex to the NPC would be affected in this mutant. Thus, different organisms may regulate this Y–Nup82 complex interaction via different motifs or mechanisms.

Regarding the physiological role of the interaction between *C. thermophilum* Y- and Nup82 complexes mediated by a SLiM in Nup145C, this might be also relevant during the cell cycle or embryonic development depending on the organism. *Drosophila* blastoderm embryos for instance contain large amounts of 'precursor' NPCs, stored in annulate lamellae, lacking both the Nup214/88 complex as well as the Nup62 complex (homologs of the Nup82 and Nsp1 complexes, respectively)[47]. Filamentous fungi undergo semi-closed mitosis in which the nuclear envelope and parts of the NPC scaffold are preserved, but several Nups dissociate during mitosis, including the Nup82 complex[48, 49]. Vertebrates entirely break down their nuclear envelope and NPCs

during mitosis[50]. Thus, in both vertebrates and filamentous fungi, the mode of association of the Nup82 complex with the NPC scaffold has to be mitotically regulated, which could occur by phosphorylation[48, 51]. Interesting in this context is that the NTD of human Nup96 (yeast Nup145C) is mitotically phosphorylated (see also Supplementary Fig. 2), which might contribute to the detachment of the peripheral NPC structures or even NPC disassembly[6, 51, 52]. The situation in yeast is not known. Yeast undergoes closed mitosis, that is, chromosomal separation occurs within the nuclear envelope and NPCs remain fully assembled throughout the cell cycle. It will be interesting to discover whether a SLiM in the yeast Nup145C N-terminal extension—one that functions differently—exists, albeit perhaps one not regulated by phosphorylation.

Although the *Ct*Nup145C-B motif and the following DIM within *Ct*Nup145C are located directly adjacent to each other, they are able to bind simultaneously to the *Ct*Nup82 complex and *Ct*Sec13. With knowledge of the crystal structure of the *Sc*Nup84 complex, it is plausible that the integrating *Ct*Sec13 at the DIM of *Ct*Nup145C helps to expose the *Ct*Nup145C-B motif for efficient 'grappling' of the *Ct*Nup82 complex. Because the Nup82 complex localizes exclusively on the cytoplasmic face of the NPC, the question arises if the Nup145C-B motif might have a function in the NR. Interestingly, deletion of Nup145C in yeast leads to a cytoplasmic mislocalization of the nuclear basket proteins Mlp1 and Mlp2[27]. However, also various other asymmetrically distributed NPC components, such as in example Nup100, Nup116, Nup145N, and Nup60 could in principle contribute to establishing a directionality cue across the nuclear envelope and must be further explored in the context of NPC assembly in the future. Obviously, symmetric tethering sites such as the Nup145C-B motif must either have compartment-specific binding partners or remain unoccupied on one of the two faces of the NPC.

Finally, we discovered that the short *Ct*Nup145C-B SLiM can bind only to the assembled *Ct*Nup82 complex, but not to its individual subunits. The same has been observed for *Ct*Nic96-IM-1, a SLiM in a structural Nup that links the Nsp1–Nup49–Nup57 channel complex to the IR complex[8, 9]. Cooperative binding might guarantee that the FG repeats of the Nsp1 complex become optimally exposed towards the central transport channel[8, 9]. Also the Nup82 complex contains many FG repeats as part of the FG-repeat domain of Nup159 and Nsp1, and the Nup82–Nup159–Nsp1 complex is formed by the coiled-coil domain of Nsp1 shared with the Nsp1 complex and coiled-coil domains of Nup82 and Nup159 homologous to Nup57 and Nup49, respectively[37], suggesting an evolutionary relationship between the outer ring Nup82 complex and the central channel Nsp1 complex. Thus, we suggest that the Nup145C-B and Nic96-IM-1 SLiMs perform related functions, that is to cooperatively associate with evolutionary related subcomplexes. Further, the Nup145C and Nic96 α-solenoid domains are homologous[53, 54]. Therefore, the whole systems of Nup145C–Nup82–Nup159–Nsp1 and the Nic96–Nup57–Nup49–Nsp1 assemblies might have arisen from a common ancestor, similar to how some Nup genes have been duplicated during evolution to fulfill related, yet different roles in the NPC[8, 54, 55]. Notably, there are also differences in how the two subcomplexes interact with the scaffold. While the Nup82 complex interacts with various other components of the NPC inner and outer rings, Nic96 is thus far the only known anchor of the Nsp1 complex[8, 23].

The prevalent model describing the origin of the NPC is the proto-coatomer hypothesis, arguing that both the NPC and eukaryotic vesicle coats emerged from a common membrane-coating protein complex[56]. This hypothesis is mainly based on the predominant presence of α-solenoid and β-propeller folds in

structural Nups and vesicle coating proteins. However, it is now evident that coiled-coil interactions recruit the 'peripheral' Nsp1/Nup62 and Nup82/Nup214 complexes, which carry the majority of FG repeats and are indispensible for nucleocytoplasmic transport. We therefore propose to extend the proto-coatomer hypothesis by a second aspect that is the addition of the transport barrier function by means of cooperative coiled-coil interactions.

## Methods

**Generation of Nup overexpression plasmids.** All plasmids used in this study were either previously published[8, 15] or generated by subcloning the appropriate full-length or truncated Nup open reading frames (ORFs) from the previously published plasmids using standard PCR and molecular cloning techniques. The Nup ORFs were cloned into E. coli or S. cerevisiae overexpression vectors containing various combinations of N- and/or C-terminal affinity tags, including GST, protein A (ProtA), polyhistidine (His), and Flag. In some cases, affinity tags were fused to tobacco etch virus (TEV) cleavage sites in order that the tags could be removed. All plasmids used in this study are listed in Supplementary Table 1. Plasmids were transformed into E. coli or yeast strains using standard protocols.

**Overexpression of Nups in E. coli and S. cerevisiae.** E. coli expression vectors were transformed into E. coli strain BL21 CodonPlus. E. coli cells expressing CtNups were grown in lysogeny broth medium at 37 °C to an $OD_{600 \, nm}$ value of 0.4. Cells were switched to a 23 °C environment for 30 min before expression was induced with 0.5 mM isopropyl β-D-1-thiogalactopyranoside (IPTG) for 2 h at 23 °C. E. coli cells expressing ScNups were grown in minimal medium at 37 °C to an $OD_{600 \, nm}$ value of 0.4. Cells were switched to a 16 °C environment for 30 min before expression was induced with 0.5 mM (IPTG) for 16 h at 16 °C.

S. cerevisiae expression vectors were transformed into the wild-type yeast strain Ds1-2b[57]. Transformants harboring these plasmids were first grown at 30 °C overnight in the appropriate synthetic dextrose complete (SDC) drop-out medium. The SDC pre-cultures were subsequently diluted to 1.25% in yeast extract peptone galactose medium (YPG, containing 1% (w/v) yeast extract, 2% (w/v) bacto-peptone, and 2% (w/v) galactose) and grown at 30 °C overnight to an $OD_{600 \, nm}$ value of 4.5 to induce the GAL promoters.

**Protein purification.** All steps were carried out at 4 °C unless otherwise stated. Harvested E. coli and yeast cells were lysed in HEPES–NB buffer ('Normal Buffer', 20 mM HEPES, pH 7.5, 150 mM NaCl, 50 mM KOAc, 2 mM Mg(OAc)$_2$, 5% glycerol and 0.01% (v/v) NP40) supplemented with SigmaFast protease inhibitor cocktail tablets (Sigma–Aldrich). E. coli cells were lysed using a microfluidizer (Microfluidics 110 L). Yeast cells were lysed by cryogenic grinding (MM 400, Retsch). Lysates were cleared by centrifugation at 35,000 g for 25 min at 4 °C.

Overexpressed Nups were purified from lysates using commercially available affinity beads. ProtA-, GST-, and Flag-tagged proteins were purified by incubation with IgG beads (IgG Sepharose 6 Fast Flow, GE Healthcare), GSH beads (Protino Glutathione Agarose 4B, Macherey–Nagel), and anti-Flag beads (Anti-Flag M1 Agarose Affinity Gel, Sigma–Aldrich), respectively, at 4 °C for 60 min or overnight. Beads were washed with HEPES–NB buffer and, if necessary, eluted by incubation with His-tagged TEV protease in HEPES–NB buffer supplemented with 1 mM dithiothreitol for 60 min at 16 °C or overnight at 4 °C. Flag-tagged proteins were alternatively eluted by incubation in HEPES–NB buffer containing Flag peptide (Sigma–Aldrich) for 60 min at 4 °C. Lysates containing His-tagged Nups were supplemented with 20 mM imidazole and applied to Ni–NTA beads (His-Select HF Nickel Affinity Gel, Sigma–Aldrich) by gravity flow at 4 °C. Beads were washed with HEPES–NB buffer containing 20 mM imidazole and bound proteins were eluted with HEPES–NB buffer containing 500 mM imidazole. GST-TEV-CtNup145C-A-Flag and GST-TEV-CtNup145C-B-Flag (Fig. 4b) were first purified via GSH beads, washed, TEV eluted, immobilized on anti-Flag beads, and washed again. When used as prey proteins, GST-TEV-CtNup145C-A-His and GST-TEV-CtNup145C-B-His were purified in a single Ni–NTA affinity step. In this case, the GST-TEV tags were not used for affinity purification but rather to increase the molecular weight of the protein domain/motif to facilitate detection by sodium dodecyl sulfate polyacrylamide gel electrophoresis (SDS-PAGE) and coomassie staining. The ScY-vertex was purified via GSH beads from an E. coli strain co-expressing plasmids pPROEXHT-GST-TEV-ScNup145C-ScSec13-T7-ScNup120 and pET24d-ScNup85-ScSeh1. The ScNup82 complex was tandem affinity purified from Ds1-2b yeast co-expressing plasmid YEplac195-P2-ScDyn2-P1-ScNUP159-ΔFG with either YEplac181-P2-ScNsp1C-P1-ScNup82-Flag-TEV-ProtA (Supplementary Fig. 1, lane 1) or YEplac181-P2-ScNsp1C-Flag-TEV-ProtA-P1-ScNup82 (Supplementary Fig. 1, lane 2). In both cases, the ScNup82 complex was first purified via IgG beads, eluted with TEV protease, purified with anti-Flag beads, and finally eluted using Flag peptide (details see above).

**In vitro reconstitution of the CtNup82–Y complex.** The CtY-vertex (CtNup145C–Nup85–Nup120) was purified from Ds1-2b yeast co-expressing plasmids YEplac181-P2-CtNup120-P1-ProtA-TEV-His-CtNup85 and YEplac112-

CtNup145C using IgG beads (details see previous paragraph). Beads were washed and the CtY-vertex was eluted with TEV protease. The CtNup82 complex was first purified from Ds1-2b yeast co-expressing plasmids YEplac195-CtNup82, YEplac112-CtNsp1C-His, and YEplac181-Flag-CtNup159C using Ni–NTA beads as described above. The imidazole-eluted CtNup82 complex was then immobilized on anti-Flag beads, washed, and finally incubated with an approximately threefold molar excess of CtY-vertex for 45 min at 16 °C. Subsequently, the anti-Flag beads were washed with excess HEPES–NB buffer and the CtNup82–Y complex (CtNup82–Nup159C–Nsp1C–Nup145C–Nup85–Nup120) was eluted with the Flag peptide in HEPES–NB buffer lacking NP40 and submitted to XL-MS.

**Crosslinking mass spectrometry.** The in vitro assembled CtNup82–Y complex was split into two batches of ~80 µg each (~0.5 µg µl$^{-1}$). Samples were crosslinked by incubation with 0.5 or 2 mM H$_{12}$/D$_{12}$ isotope-coded disuccinimidyl suberate (Creative Molecules) at 37 °C for 30 min. Quenching, proteolytic digestion, and acquisition and analysis of MS data was carried out as previously described[6, 10]. In short, the reaction was quenched by addition of 50 mM ammonium bicarbonate for 10 min at 37 °C. Crosslinked proteins were denatured using urea and Rapigest (Waters) at a final concentration of 4 M and 0.05% (w/v), respectively. Reduction was performed with 10 mM DTT (30 min at 37 °C) and carbamidomethylation of cysteins with 15 mM iodoacetamide (30 min in the dark). Proteins were digested with 1:100 (w/w) LysC (Wako Chemicals GmbH, Neuss, Germany) for 4 h at 37 °C in a first step. Second, the urea concentration was diluted to 1.5 M and the digestion was finalized with 1:50 (w/w) trypsin (Promega GmbH, Mannheim, Germany) overnight at 37 °C. Samples were then acidified with 10% (v/v) TFA and desalted using MicroSpin columns (Harvard Apparatus) following standard procedures. Crosslinked peptides were enriched using size exclusion chromatography (SEC) using a Superdex Peptide PC 3.2/30 column on an Ettan LC system (GE Healthcare) at a flow rate of 50 µl min$^{-1}$. Fractions eluting between 0.9 and 1.4 ml were evaporated to dryness and reconstituted in 20–50 µl 5% (v/v) ACN in 0.1% (v/v) FA according to 215 nm absorbance. Between 2 and 10% of the collected fractions were analyzed in duplicates by liquid chromatography–mass spectrometry (LC-MS)/MS using a nanoAcquity UPLC system (Waters) connected online to LTQ-Orbitrap Velos Pro instrument (Thermo). The resulting raw files were converted to centroid mzXML using the Mass Matrix file converter tool. Data analysis was performed using xQuest and xProphet[58] searching against a fasta database containing the sequences of the crosslinked proteins. For further analysis, only crosslinks with an xQuest linear discriminant (ld) score[58] of at least 25 and an estimated false discovery rate[59] lower than 5% were used (Supplementary Table 2), in line with previous XL-MS analyses of known protein structures[59, 60]. The entire data set containing all crosslinks regardless of their ld-score is shown in Supplementary Data 1. The xiNET crosslink viewer[61] was used to create the crosslink map shown in Fig. 1b. The crosslinks were mapped to the model and analyzed using Xlink Analyzer[60].

**In vitro binding assays.** Purified prey Nups were added in an approximately fivefold molar excess over bait Nups or bait Nup complexes immobilized via ProtA, GST, or Flag tags in the presence of E. coli lysate to compete for nonspecific binding. After incubation for 50 min at 16 °C, beads were washed with excess HEPES–NB buffer and bound proteins were eluted in SDS sample buffer by incubation for 2 min at 90 °C (60 °C for ProtA-tagged bait Nups). Eluates were analyzed by SDS-PAGE and coomassie staining.

**Size exclusion chromatography.** SEC was performed at 4 °C using a Superdex 200 10/300 GL column attached to the ÅKTA Basic system (GE Healthcare). HEPES–NB buffer was used at a flow rate of 0.4 ml min$^{-1}$. Fractions were analyzed by SDS-PAGE and coomassie staining. In the case of the CtNup82 complex + CtNup145C-B run, fractions were first precipitated with a final concentration of 15% (v/v) trichloroacetic acid before protein pellets were washed with cold acetone, dissolved in SDS sample buffer, and analyzed by SDS-PAGE.

**Homology modeling.** The model of the CtNup82 complex was built by homology modeling based on the structure of the CtNsp1 complex (CtNsp1C–Nup49C–Nup57C heterotrimer bound to the IM-1 of CtNic96, PDB ID: 5cws[9]). CtNup49 was used a template for modeling CtNup159, and CtNup57 as a template for CtNup82. The sequence alignments for modeling were generated using the HHPRED server[62] and refined manually based on secondary structure predictions from the GeneSilico MetaServer[63]. The atomic model was generated using Modeller[64].

**Data availability.** The authors declare that the data supporting the findings of this study are available within the paper and its Supplementary Information files, and are available from the corresponding author upon request. The mass spectrometry proteomics data have been deposited to the ProteomeXchange Consortium via the PRIDE[65] partner repository with the data set identifier PXD007043.

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

## Acknowledgements

Plasmids pET24b-GST-TEV-*Ct*Nup159C-Flag, YEplac112-ProtA-TEV-*Ct*Nup82-Flag, and pET24b-GST-TEV-*Ct*Nsp1C-Flag were kindly provided by Jessica Fischer (Hurt lab). Plasmid pPROEXHT-GST-TEV-*Sc*Nup145C-*Sc*Sec13-T7-*Sc*Nup120 was kindly provided by Malik Lutzmann (Hurt lab). This work was supported by grants from the European Research Council (grant 309271-NPCAtlas to M.B), the Deutsche For-schungsgemeinschaft (DFG Hu363/13-1 to E.H.), and the EMBL Interdisciplinary Postdoc Programme under Marie Curie COFUND actions (J.K.).

## Author contributions

R.T. designed and performed all in vitro reconstitution experiments and the truncation analysis. A.v.A. performed XL-MS and analyzed data. J.K. analyzed data and performed homology modeling. M.B. analyzed data and oversaw the project. E.H. analyzed data, directed the project, and together with R.T. wrote the manuscript.

## Additional information

**Competing interests:** The authors declare no competing financial interests.

