## [Peer Review File · Nature Communications]

Reviewers' comments:

Reviewer #1 (Remarks to the Author):

Elucidating the structure of the NPC remains an important topic in structural cell biology. While we possibly have a fairly complete description of the human NPC, decidedly due to the work conducted in the corresponding authors' respective labs, the situation for single-cell organisms like *C. thermophilum* is not as clear. It is not known to what extent the working model for the human NPC is applicable to the fungal NPC, which seems to be much smaller and possibly contains much fewer copies of nucleoporins. Therefore, it is important to elucidate the subcomplex interactions for *C. thermophilum*, or other fungi, and compare and contrast with the findings in humans and other metazoa.

The authors show with a combination of XL-MS, pull-downs, and modelling that the unstructured N-terminal region of Nup145C, is necessary and sufficient to recruit the trimeric Nup82 complex to the Y complex. They take this to claim that Nup145C recruits the Nup82 complex to the cytoplasmic face of the NPC.

Comments:

1.) Fernandez-Martinez et al. (Cell 2016, PMID:27839866) recently showed that in *S. cerevisiae* the trimeric Nup82 complex is primarily recruited via Nup85, rather than Nup145C, to the Y complex. While this paper is mentioned, the apparent conflict with the presented data is not well addressed. I think that the authors need to be much more thorough in comparing and contrasting the studies. Do they think that Ct and Sc are fundamentally different with regard to Nup82 complex recruitment? Or do they think that the Fernandez-Martinez paper is flawed for some reason? XL-MS is used in both studies, so it seems surprising to see such differences. This is a major point.

2.) Given the fact that there are at least a number of other interactions between the Nup82 complex and the scaffold of the NPC (i.e., ctNup145N to ctNup170 and ctNup192, Fischer et al., NSMB 2015; Lin et al., Science2016) Nup145C does not appear to be the only anchor. The comparison with Nic96, so far the exclusive anchor of the central Nsp1 complex to the NPC scaffold is therefore a stretch. In this light, I find the title, the abstract, and the last part of the discussion much too generalizing. I would argue that the recruitment of the Nsp1 complex versus the Nup82 complex is substantially different, even though they do share a number of commonalities, for example the protein Nsp1 itself, which has been known for a long time. I

wish this would be addressed, rather than glossed over for the sake of generating a cute story line.

Minor Comments:

- 1.) In Figure 2C, Nup 145C-NTD is unexpectedly shown to pull down Sec13 (lane 12), while this interaction does not register in the presence of the Nup82complex (lane 14). This warrants an explanation.
- 2.) In Figure 3, there is a significant contrast between Sec13 being pulled down with Y-vertex (lane 14), but surprisingly poorly with Y-vertex-CTD (lane 15). I would have expected equal efficiency, comparing the result with Figure 2C, lanes 6+10.
- 3.) Given the fact that there may be a difference between Nup82 complex recruitment in Ct vs. Sc, it is surprising that the authors chose to use the Sc Y complex crystal structure for comparison (Figure 2a) rather than the much better resolved, Y-vertex structure from *M. thermophila*, a very close relative to Ct (PDB code 4Y CZ).
- 4.) I am surprised that the authors speculate that an Mlp1 or Mlp2 homodimeric coiled-coil may engage with Nup145C-B similar to the trimeric coiled-coil that Nsp1-Nup82-Nup159 form (page 11). If the trimeric coil is argued to be necessary for Nup145C binding, then the engagement with a dimeric coiled-coil may be very different. Coiled-coils are so widespread and diverse that it is far-fetched to suggest common assembly principles just because of the presence of a coiled-coil element.
- 5.) Please replace reference 53 with the more appropriate papers: Devos et al, PNAS 2006; and Brohawn et al., Science 2008).
- 6.) While introducing the differences among Y complexes of different origin (page 3), the authors should include to mention that Seh1 is absent in thermophilic ascomycota, referencing ref. 15 and Kelley et al, NSMB 2015.

Reviewer #2 (Remarks to the Author):

Nuclear pore complexes (NPCs) feature a symmetric core structure as well as asymmetric cytoplasmic and nuclear modules. The study by Teimer et al. now studied one of these asymmetric interactions, namely the anchorage of the cytoplasmic Nup159-Nsp1-Nup82 complex to the NPC. It was known before that this Nup82 complex binds the Y-complex. The current study elucidates a binding motif in the Y-complex component Nup145C by means of crosslinking/ mass spectrometry as well by reconstitution experiments for the interaction. Overall, this is a nice paper, well written, though one could also argue that the findings are a bit incremental.

Specific points:

1. The study gives no satisfactory explanation as to how a symmetrical scaffold can provide asymmetric binding sites. It seems likely that important binding partners of the Nup82 complex that contribute to its asymmetric tethering are yet to be identified. For the sake of transparency, this should be spelled out clearly.

2. The interaction seems weak in nature, since not always stoichiometric complexes could be formed (most evident in Figure 3). In support of this the authors themselves raise the possibility of additional interactions with the Y-vertex. A gel filtration analysis of the reconstituted complexes (Nup82 complex-Nup145C/Nup145C-B) would give more insights into the strength of this interaction.

3. In Figure 3 the faint Nup145C-CTD staining appears equally strong in lanes 9, 11, 13, 15. This observation actually argues against the author's suggestion, that other members of the Y-complex (e.g. Nup85) would contribute to the overall binding of the Nup82 complex. Instead, Nup145C-CTD might either display some unspecific background binding to the beads or provides one or more (very weak) interactions sites, independent of the identified SLIM. To distinguish between both, the authors need to test the first possibility by adding an empty bead control.

4. It would be highly interesting to know if deleting or mutating the identified SLIM motif in scNup145C would block incorporation of the Nup82 complex into yeast NPCs in vivo.

5. A more informative/ explicit naming of the binding motif (right now called 'Nup145C-B') would add clarity.

6. Is the statement that the alpha solenoid domains in Nup145C and Nic96 being homologous really a hard fact? This argument is then used to further argue that the Nic96-Nsp1 complex and Nup145C-Nup82 complex interactions would also be homologous. This is confusing, because it would imply that the alpha solenoid domains account for these interactions, which is not the case.

Reviewer #1 (Remarks to the Author):

Elucidating the structure of the NPC remains an important topic in structural cell biology. While we possibly have a fairly complete description of the human NPC, decidedly due to the work conducted in the corresponding authors' respective labs, the situation for single-cell organisms like *C. thermophilum* is not as clear. It is not known to what extent the working model for the human NPC is applicable to the fungal NPC, which seems to be much smaller and possibly contains much fewer copies of nucleoporins. Therefore, it is important to elucidate the subcomplex interactions for *C. thermophilum*, or other fungi, and compare and contrast with the findings in humans and other metazoa. The authors show with a combination of XL-MS, pull-downs, and modelling that the unstructured N-terminal region of Nup145C, is necessary and sufficient to recruit the trimeric Nup82 complex to the Y complex. They take this to claim that Nup145C recruits the Nup82 complex to the cytoplasmic face of the NPC.

Comments: 1.) Fernandez-Martinez et al. (Cell 2016, PMID:27839866) recently showed that in *S. cerevisiae* the trimeric Nup82 complex is primarily recruited via Nup85, rather than Nup145C, to the Y complex. While this paper is mentioned, the apparent conflict with the presented data is not well addressed. I think that the authors need to be much more thorough in comparing and contrasting the studies. Do they think that Ct and Sc are fundamentally different with regard to Nup82 complex recruitment? Or do they think that the Fernandez-Martinez paper is flawed for some reason? XL-MS is used in both studies, so it seems surprising to see such differences. This is a major point.

We agree that this is worth clarifying. We think that *C. thermophilum* and *S. cerevisiae* Nup82 complexes may arrange in similar orientation relatively to the Y-complex, but the contribution of particular interaction sites to the binding affinity is different.

First, we have performed *in vitro* reconstitution studies using recombinant yeast Y-complex and yeast Nup82 complex under similar *in vitro* binding conditions as compared to the CtNups. However, we do not find that the yeast Y-complex and Nup82 complex form a stable assembly. We show this new data in **Supplementary Fig. 1**. Since we found no interaction, we did not proceed to test ScNup145C or ScNup85 individually for binding towards the ScNup82 complex. Importantly, we have observed in the past that interactions between CtNups can be more easily reconstituted than using the orthologous ScNups (see e.g. Fischer et al., NSMB, 2015). It should also be emphasized that Fernandez-Martinez et al. neither showed direct biochemical (*in vitro* reconstitution) data of the ScY-Nup82-complex nor that ScNup85 is sufficient to recruit the ScNup82 complex *in vitro*. Instead, they have affinity-purified ScNup84 complex from yeast cells under conditions that brings some ScNup82 along and analyzed this preparation by XL-MS. They identified three crosslinks between ScNup85 and the ScNup82 complex. Notably, such data demonstrate proximity not direct biochemical interaction. The modeling by Fernandez-Martinez et al was done without considering physical potential or an EM data covering both complexes but based on these three crosslinks and crosslinks connecting other subunits than ScNup85 - and as such - placed the ScY- and ScNup82 complex respectively to each other.

Second, our XL-MS data does not exclude proximity with Nup85. In fact, we also observed one crosslink from the N-terminus of CtNup85 to the CtNup154C-B motif, which suggests that the N-terminus of Nup85 is in vicinity to the Nup154C-B-Nup82 interaction region. The three crosslinks observed by Fernandez-Martinez et al. link the N-terminus of Nup85 with the C-terminal part of ScNup82 (coiled-coil domain), which is the same part that we found to interact with the Nup154C-B motif in *C. thermophilum*. Moreover, the N-terminus of ScNup154C crosslinked to the coiled-coil regions of Nup82 and Nup159. Thus, both XL-MS data sets suggest proximity of

the N-terminus of Nup85, Nup82-Nup159-Nsp1 coiled-coils, and the N-terminus of Nup145C. In conclusion, we do not think that both studies contradict each other.

Third, we have performed a deletion experiment in yeast cells. Deletion of the entire N-terminal domain from yeast Nup145C, which includes hypothetical SLiMs, did not cause any growth defect in a yeast Nup145 null mutant in the presence of plasmid-borne Nup145N (R.T. and E.H., unpublished data, see reviewer #2, comment 4).

Based on all these findings, we have included the following paragraph into the discussion section to bring this across more clearly:

“The interaction between Y- and Nup82 complex in *C. thermophilum* is consistent with the recent, aforementioned study done in yeast implying that another member of the Y-complex, Nup85, significantly contributes to the overall binding of the Nup82 complex³⁷. This notion was based on that Nup85 and Nup145C crosslink to members of the Nup82 complex and that truncations of the Nup85/Seh1 arm affect the incorporation of Nup82 into the NPC. However, no *in vitro* reconstitution was performed or stoichiometric binding observed. Since our *in vitro* reconstitution studies with yeast Y- and Nup82 complexes were negative, we cannot further address the question of how yeast Y-Nups directly contribute to binding to the yeast Nup82 complex, as suggested by Fernandez-Martinez and coworkers³⁷. Notably, the SLiM as found in CtNup145C is either absent or has a divergent sequence in yeast Nup145C (Supplementary Fig. 2b) suggesting that the ScNup145C could contribute less to this interaction between the yeast Y- and Nup82 complexes. Consistent with this data, deletion of the entire N-terminal domain from yeast Nup145C, which includes hypothetical SLiMs, did not cause any growth defect in a yeast *nup145* null mutant in the presence of plasmid-borne Nup145N (R.T. and E.H., unpublished data), making it unlikely that the targeting of the yeast Nup82 complex to the NPC would be affected in this mutant. Thus, different organisms may regulate this Y–Nup82 complex interaction via different motifs or mechanisms.”

2.) Given the fact that there are at least a number of other interactions between the Nup82 complex and the scaffold of the NPC (i.e., ctNup145N to ctNup170 and ctNup192, Fischer et al., NSMB 2015; Lin et al., Science2016), Nup145C does not appear to be the only anchor. The comparison with Nic96, so far the exclusive anchor of the central Nsp1 complex to the NPC scaffold is therefore a stretch. In this light, I find the title, the abstract, and the last part of the discussion much too generalizing. I would argue that the recruitment of the Nsp1 complex versus the Nup82 complex is substantially different, even though they do share a number of commonalities, for example the protein Nsp1 itself, which has been known for a long time. I wish this would be addressed, rather than glossed over for the sake of generating a cute story line.

Agreed. We have the following passage into the discussion section to make this more transparent: “Notably, there are also differences in how the two subcomplexes interact with the scaffold. While the Nup82 complex interacts with various other components of the NPC inner and outer rings, Nic96 is thus far the only known anchor of the Nsp1 complex^{8,23}.”

Moreover, we have also revised the title (which is now more specific and less generalizing) and abstract.

Minor Comments:

1.) In Figure 2C, Nup 145C-NTD is unexpectedly shown to pull down Sec13 (lane 12), while this interaction does not register in the presence of the Nup82complex (lane 14). This warrants an explanation.

In a number of *in vitro* binding assays performed in the lab, we observed that CtSec13 exhibits some background binding to other Nups in the absence of its binding partner CtNup145C-DIM (unpublished results). We assume that CtSec13 adopts an incomplete β -propeller fold in the absence of CtNup145C-DIM, which could explain this unspecific binding. Notably, CtNup145C-NTD recruited substantially lower amounts of CtSec13 (Figure 2c, lane 12) than full-length CtNup145C (lane 4) and -CTD (lane 8). Furthermore, as noted by this reviewer, when adding CtNup82 complex, the CtNup145C-NTD-Sec13 interaction was disrupted and a CtNup145C-NTD-Nup82-Nup159C-Nsp1C complex was formed (Figure 2c, lane 14). In contrast, the interaction of CtSec13 to full-length CtNup145C and CtNup145C-CTD was not disturbed by the presence of the CtNup82 complex (lanes 6 and 10). We therefore suggest that the CtNup145C-NTD-Sec13 interaction is not relevant *in vivo* as Sec13 is recruited to the Y-complex by Nup145C-DIM (see e.g. Hsia, Cell, 2007; Kelley, NSMB, 2015; Stuwe, Science, 2015), which is located at the start of our CTD construct. A short summary of these findings has now been included into the revised manuscript.

2.) In Figure 3, there is a significant contrast between Sec13 being pulled down with Y-vertex (lane 14), but surprisingly poorly with Y-vertex-CTD (lane 15). I would have expected equal efficiency, comparing the result with Figure 2C, lanes 6+10.

We do not find this difference between lanes 14 and 15 in Figure 3 so surprising for the following reason: It is true that Fig. 2C, lane 10 shows that CtNup145C-CTD is sufficient to pull down CtSec13. However, since CtSec13 is smaller than CtNup145C-CTD, the intensity of the CtSec13 band is considerably lower than that of CtNup145C-CTD. In Figure 3, it is important to note that the CtNup82 bait complex pulled down only trace amounts of CtNup145C-CTD (lanes 9, 11, 13, 15), in contrast to near-stoichiometric amounts of full-length CtNup145C (lanes 8, 10, 12, 14). We expect these trace amounts of CtNup145C-CTD to recruit amounts of CtSec13 that are below the detection limit of the Coomassie stain.

3.) Given the fact that there may be a difference between Nup82 complex recruitment in Ct vs. Sc, it is surprising that the authors chose to use the Sc Y complex crystal structure for comparison (Figure 2a) rather than the much better resolved, Y-vertex structure from *M. thermophila*, a very close relative to Ct (PDB code 4YCZ).

While it is true that *M. thermophila* is a closer relative to *Ct*, our reasoning was as follows: for the crystallization of the *M. thermophila* Y complex (Kelley et al.), a construct was used, where MtSec13 was fused N-terminally to MtNup145C and 232 N-terminal residues of MtNup145C were excluded from the construct for crystallization. The N-terminal region of Nup145C, upstream of the domain invasion motif, was however of great interest to our work since in CtNup145C the corresponding region contains the Nup82 interaction motif identified in our work. For the yeast structure shown in Fig. 2a (Stuwe et al, PDB ID: 4xmm), a longer ScNup145C construct was used that (i) included most of the N-terminal domain and (ii) was not artificially fused to Sec13. Although most of the N-terminal region remains unresolved in the crystal structure, the resolved fragments and inspection of unassigned densities (data not shown) suggest a possible location of this region at

the Y-vertex. We therefore have chosen the Sc structure to depict the possible location of the N-terminal residues of Nup145C.

4.) I am surprised that the authors speculate that an Mlp1 or Mlp2 homodimeric coiled-coil may engage with Nup145C-B similar to the trimeric coiled-coil that Nsp1-Nup82-Nup159 form (page 11). If the trimeric coil is argued to be necessary for Nup145C binding, then the engagement with a dimeric coiled-coil may be very different. Coiled-coils are so widespread and diverse that it is far-fetched to suggest common assembly principles just because of the presence of a coiled-coil element.

We agree and have removed this statement.

5.) Please replace reference 53 with the more appropriate papers: Devos et al, PNAS 2006; and Brohawn et al., Science 2008).

We have changed the references accordingly.

6.) While introducing the differences among Y complexes of different origin (page 3), the authors should include to mention that Seh1 is absent in thermophilic ascomycota, referencing ref. 15 and Kelley et al, NSMB 2015.

We have included this information into the revised manuscript.

Reviewer #2 (Remarks to the Author):

Nuclear pore complexes (NPCs) feature a symmetric core structure as well as asymmetric cytoplasmic and nuclear modules. The study by Teimer et al. now studied one of these asymmetric interactions, namely the anchorage of the cytoplasmic Nup159-Nsp1-Nup82 complex to the NPC. It was known before that this Nup82 complex binds the Y-complex. The current study elucidates a binding motif in the Y-complex component Nup145C by means of crosslinking/ mass spectrometry as well by reconstitution experiments for the interaction. Overall, this is a nice paper, well written, though one could also argue that the findings are a bit incremental.

Specific points:

1. The study gives no satisfactory explanation as to how a symmetrical scaffold can provide asymmetric binding sites. It seems likely that important binding partners of the Nup82 complex that contribute to its asymmetric tethering are yet to be identified. For the sake of transparency, this should be spelled out clearly.

It was not our goal of this study to explain the asymmetry, but for clarification, we have included the following passage into the discussion: *“Because the Nup82 complex localizes exclusively on the cytoplasmic face of the NPC, the question arises if the Nup145C-B motif might have a function in the nuclear ring. Interestingly, deletion of Nup145C in yeast leads to a cytoplasmic mislocalization of the nuclear basket proteins Mlp1 and Mlp2 (ref. 27). However, also various other asymmetrically distributed NPC components, such as in example Nup100, Nup116, Nup145N and Nup60 could in principle contribute to establishing a directionality cue across the nuclear envelope and must be further explored in the context of NPC assembly in the future. Obviously, symmetric tethering sites such as the Nup145C-B motif must either have compartment-specific binding partners or remain unoccupied on one of the two faces of the NPC.”*

2. The interaction seems weak in nature, since not always stoichiometric complexes could be formed (most evident in Figure 3). In support of this the authors themselves raise the possibility of additional interactions with the Y-vertex. A gel filtration analysis of the reconstituted complexes (Nup82 complex-Nup145C/Nup145C-B) would give more insights into the strength of this interaction.

To address this question we have reconstituted C τ Nup82–Nup159C–Nsp1C complex with the Nup145C-B motif and performed gel filtration chromatography (revised Supplementary Fig. 3). Apparently, the C τ Nup145C-B motif co-eluted with the C τ Nup82 complex. These results indicate that the interaction between SLiM C τ Nup145C-B and the C τ Nup82 complex is rather robust, which is mentioned in the Results section of the revised manuscript.

3. In Figure 3 the faint Nup145C-CTD staining appears equally strong in lanes 9, 11, 13, 15. This observation actually argues against the author's suggestion, that other members of the Y-complex (e.g. Nup85) would contribute to the overall binding of the Nup82 complex. Instead, Nup145C-CTD might either display some unspecific background binding to the beads or provides one or more (very weak) interactions sites, independent of the identified SLiM. To distinguish between both, the authors need to test the first possibility by adding an empty bead control.

We now show an empty bead control, which was performed in parallel to the binding assay, revealing that C τ Nup145C-CTD does not exhibit unspecific binding to beads (Figure 3, right panel). As noted by this reviewer, this indicates that C τ Nup145-CTD may interact weakly with the C τ Nup82 complex, independently of the SLiM C τ Nup145-B. We have added this information to the Results section of the revised manuscript.

4. It would be highly interesting to know if deleting or mutating the identified SLiM motif in scNup145C would block incorporation of the Nup82 complex into yeast NPCs *in vivo*.

Although, we could not find a biochemical evidence for a stable *in vitro* interaction between yeast Y-complex and Nup82 complex (see reviewer #1, comment 1), we have performed this deletion experiment in yeast cells, requested by this reviewer. However, deletion of the entire N-terminal domain from yeast Nup145C, which includes hypothetical SLiMs, did not cause any growth defect in a yeast Nup145 null mutant in the presence of plasmid-borne Nup145N (R.T. and E.H., unpublished data; this data however is shown to this reviewer, see below). Thus, it is unlikely that the targeting of the yeast Nup82 complex to the NPC would be affected in this mutant.

Figure for the reviewer, showing this yeast genetic experiment.

5. A more informative/ explicit naming of the binding motif (right now called 'Nup145C-B') would add clarity.

A functional name, "Nup82-Complex Interaction Motif" or "82CIM", has now been suggested for the Nup145C-B motif, and used in the revised manuscript (e.g. revised Fig. 4a).

6. Is the statement that the alpha solenoid domains in Nup145C and Nic96 being homologous really a hard fact?

The homology of α -solenoid domains in Nup145C and Nic96 is strongly supported by high similarity of their structures (Devos, PLOS Biol, 2004; Brohawn, Science, 2008; Whittle, J Biol Chem, 2009) and low but significant similarity detectable at the sequence level (Promponas, Sci Rep. 2016; and HHpred probability score: 86.63, unpublished data). Although a homology of any protein domains in general is difficult to ultimately prove as a hard fact, the similarity at both structural and sequence level

suggests a divergent evolution from a common ancestor (a homology) rather than convergent evolution from different ancestors.

This argument is then used to further argue that the Nic96-Nsp1 complex and Nup145C-Nup82 complex interactions would also be homologous. This is confusing, because it would imply that the alpha solenoid domains account for these interactions, which is not the case.

We have not actually meant that the homology between α -solenoid domains in Nup145C and Nic96 implies a homology of Nic96-Nsp1 complex and Nup145C-Nup82 complex interactions, but rather that it implies that “the whole systems of Nup145C–Nup82–Nup159–Nsp1 and the Nic96–Nsp1–Nup49–Nup57 assemblies might have arisen from a common ancestor”. This is supported by their key common feature: only the fully assembled complex binds to the NPC scaffold. Since a homology between CfNup145C-B motif and conserved motifs in ScNup145C cannot be supported due to low sequence similarity, we do not state that the interaction interfaces between Nic96-Nsp1 complex and Nup145C-Nup82 are homologous. This does not change the fact that the entire complexes can be homologous. We realized that the paragraph indeed had a confusing logic in the initial version of the manuscript and revised it as follows: “Thus, we suggest that the Nup145C-B and Nic96-IM-1 SLiMs perform related functions, that is to cooperatively associate with evolutionary related subcomplexes. Further, the Nup145C and Nic96 α -solenoid domains are homologous^{54,55}. Therefore, the whole systems of Nup145C–Nup82–Nup159–Nsp1 and the Nic96–Nsp1–Nup49–Nup57 assemblies might have arisen from a common ancestor ...”

Reviewers' Comments:

Reviewer #1 (Remarks to the Author):

The authors have addressed my concerns adequately and I recommend publication of this revised manuscript.

Reviewer #2 (Remarks to the Author):

This is an appropriate revision and response to the reviewers' point.